# Surface Modifications of 2D-Ti$_3$C$_2$O$_2$ by Nonmetal Doping for Obtaining High Hydrogen Evolution Reaction Activity: A Computational Approach

**Fangtao Li** [1], **Xiaoxu Wang** [2],*[ID] **and Rongming Wang** [1],*

[1] Beijing Advanced Innovation Center for Materials Genome Engineering, Beijing Key Laboratory for Magneto-Photoelectrical Composite and Interface Science, School of Mathematics and Physics, University of Science and Technology Beijing, Beijing 100083, China; lft29@163.com

[2] Beijing Key Laboratory of Cloud Computing Key Technology and Application, Beijing Computing Center, Beijing Academy of Science and Technology, Beijing 100094, China

* Correspondence: wangxx@bcc.ac.cn (X.W.); rmwang@ustb.edu.cn (R.W.)

**Abstract:** As a typical two-dimensional (2D) MXene, Ti$_3$C$_2$O$_2$ has been considered as a potential material for high-performance hydrogen evolution reaction (HER) catalyst, due to its anticorrosion and hydrophilic surface. However, it is still a challenge to improve the Ti$_3$C$_2$O$_2$ surficial HER catalytic activity. In this work, we investigated the HER activity of Ti$_3$C$_2$O$_2$ after the surface was doped with S, Se, and Te by the first principles method. The results indicated that the HER activity of Ti$_3$C$_2$O$_2$ is improved after being doped with S, Se, Te because the Gibbs free energy of hydrogen adsorption ($\Delta G_H$) is increased from $-2.19$ eV to 0.08 eV. Furthermore, we also found that the $\Delta G_H$ of Ti$_3$C$_2$O$_2$ increased from 0.182 eV to 0.08 eV with the doping concentration varied from 5.5% to 16.7%. The HER catalytic activity improvement of Ti$_3$C$_2$O$_2$ is attributed to the local crystal structure distortion in catalytic active sites and Fermi level shift leads to the p-d orbital hybridization. Our results pave a new avenue for preparing a low-cost and high performance HER catalyst.

**Keywords:** Ti$_3$C$_2$O$_2$; DFT; nonmetal doping



## 1. Introduction

Environmental pollution and the energy crisis are two tough issues in human development. Therefore, it is exigent to search for green and clean energy, which not only can reduce the production of pollution gas, but is also helpful to realize the sustainable development of economy. As a green energy, hydrogen energy has been widely considered, and electrocatalytic decompose water has been treated as an effective way for hydrogen production [1–3]. Generally, Pt is the primary catalyst for hydrogen evolution reaction (HER); however, the high cost and rarity heavily limit its widespread application [4–6]. Therefore, it is desired to explore a low-cost, abundant, and outstanding catalytic activity HER catalyst [2,4–6].

Ti$_3$C$_2$X$_x$ is a two-dimensional (2D) MXene materials which evolved from bulk MAX (M = transition metals, such as Ti, V, Cr; X = C or N) [7,8]. Interestingly, Ti$_3$C$_2$O$_2$ shows excellent stability in acid conditions after being etched by hydrofluoric acid (HF). In addition, in the process of etching, the surface dangling bonds of Ti$_3$C$_2$O$_2$ can combine with the functional groups from solution. Among them, 2D-Ti$_3$C$_2$O$_2$ shows most stable structure and excellent hydrophilicity [9]. The benefit of the above merits is that 2D-Ti$_3$C$_2$O$_2$ has the potential become a high performance HER catalyst [10,11]. Unlike traditional HER catalysts, such as 2D MoS$_2$, the active sites of 2D-Ti$_3$C$_2$O$_2$ are mainly concentrated at the surface, which lead to catalytic activity lower than Pt [12–16]. Previous theoretical studies have shown that element loading or doping at the transition metal surface can effectively improve the catalytic activity, but these results need to be further verified by actual experiments [4–6]. Fortunately, we found that 2D-Ti$_3$C$_2$O$_2$ doped with other

elements has been prepared in practical experiments [17], but their HER properties still have not been studied. Doping of nonmetal elements has been proven as an effective means to regulate HER catalytic activity. Therefore, it is necessary to carry out theoretical studies on the HER catalytic activity and complex mechanism of 2D-Ti$_3$C$_2$O$_2$ after being doped with S, Se, and Te.

In this work, we carefully studied the $\Delta G_H$ of 2D-Ti$_3$C$_2$O$_2$ after being doped with S, Se, Te on the surface by first principles density functional theory (DFT). The results indicate that the HER activity of Ti$_3$C$_2$O$_2$ is effectively improved after being doped with Te. Furthermore, we also studied the effect of doping concentration on HER catalytic activity and found that the $\Delta G_H$ increased from 0.182 eV to 0.08 eV with the doping ratio varying from 5.5% to 16.7%. The improvement of HER catalytic activity is ascribed to the local crystal structure distortion of catalytic active site and Fermi level shift leading to the p-d orbital hybridization, which provides an effective theoretical model for experimental preparation.

## 2. Results and Discussion

Previous studies have demonstrated that the 2D-Ti$_3$C$_2$O$_2$ surface properties can be modified by doping S, Se, and Te; however, the doping concentration is usually low. Therefore, based on the interesting results, we took 2D-Ti$_3$C$_2$O$_2$ 3 × 3 supercell as the research model to study the HER catalytic activity of 2D-Ti$_3$C$_2$O$_2$ after being doped with X (X = S, Se, and Te) as shown in Figure 1a. As a result, there are three kinds of H adsorption sites X, O$_1$, and O$_2$ when Ti$_{27}$C$_{18}$O$_{17}$X is doped with low concentration, and two kinds of H adsorption sites, X and O$_1$, are found in high concentration Ti$_{27}$C$_{18}$O$_{17}$X$_3$.

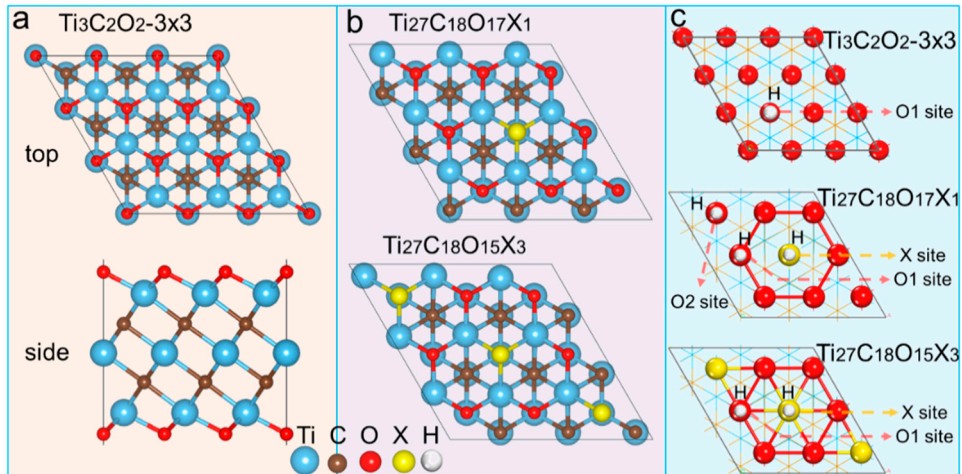

**Figure 1.** Structure diagram and $\Delta G_H$. (**a**) Two-dimensional (2D)-Ti$_2$CO$_2$ (Ti$_{27}$C$_{18}$O$_{18}$) of 3 × 3 supercell and (**b**) Ti27C18O17X and Ti$_{27}$C$_{18}$O$_{17}$X$_3$ doped with different concentrations of X (X = S, Se, and Te), and (**c**) corresponding H equivalent adsorption sites X, O1, and O2.

Figure 1b shows the $\Delta G_H$ of 2D-Ti$_3$C$_2$O$_2$ after being doped with X. It is clear that the adsorption strength of O for H was significantly changed after being X-doped, indicating that the catalytic activity was regulated. More interestingly, a new active site X appeared at the surface, which was favorable for improving the HER catalytic activity. Following this, we calculated the difference of the Gibbs free energy for adsorbed H ($\Delta G_{H*}$) according to Equation (5) to evaluate their HER activity. It should be noticed that an optimal HER activity of 2D-Ti$_3$C$_2$O$_2$ can be achieved when the absolute value of $\Delta G_{H*}$ ($|\Delta G_{H*}|$) is close to zero, meaning that the Gibbs free energy of adsorbed H is close to that of the reactant or product. Large negative $\Delta G_{H*}$ will cause bonding of adsorbed H too strong to be extracted from the catalyst surface, while more positive $\Delta G_{H*}$ will make the protons bond to the surface of catalyst too weak and difficult, both leading to slow HER kinetics. Generally, $|\Delta G_{H*}| < 0.2$ eV is a classical rule to evaluate whether a material possesses HER activity. The $\Delta G_H$ of Ti$_{27}$C$_{18}$O$_{17}$Se$_3$, Ti$_{27}$C$_{18}$O$_{17}$Te$_1$, and Ti$_{27}$C$_{18}$O$_{17}$Te$_3$ are 0.163 eV,

0.182 eV, and 0.081 eV, respectively. These comforting results manifested nonmetal doping can effectively improve the 2D-$Ti_3C_2O_2$ catalytic activity, which is also helpful for designing high-performance HER catalyst.

We added the principle of hydrogen atom adsorption on nonmetallic elements (Figure 2a) and the relationship between the catalytic activity and Bader charge transfer of doped X (Figure 2b). It was found that the decreasing electronegativity of X result in the ability of 2D-$Ti_3C_2O_2$ to obtain charge (Bader charge transfer) decreased after being doped with X. From Figure 2b, we can see that relationship between Bader charge and catalytic activity ($\Delta G_H$) was inverse proportional, and the $\Delta G_H$ value was positively correlated with the doping concentration of X. The intrinsic relationship between the catalytic activity and the doping elements provides a theoretical basis for the design and synthesis of new MXene catalysts for hydrogen evolution.

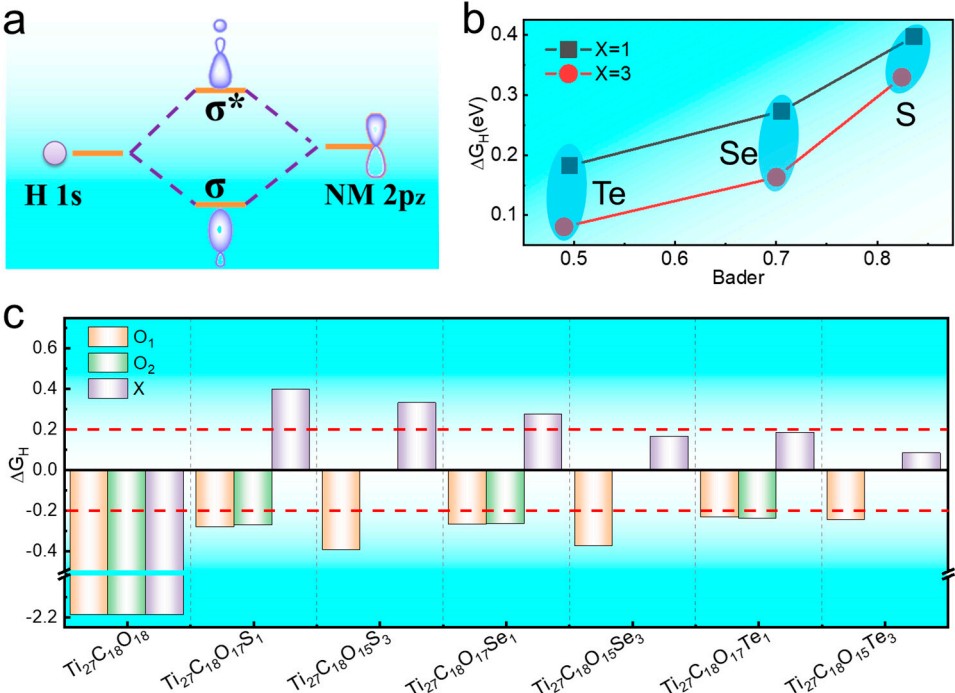

**Figure 2.** (**a**) The mechanism of hydrogen atom adsorption on nonmetal; (**b**) the relationship between catalytic activity and Bader charge transfer; (**c**) $\Delta G_H$ of 2D-$Ti_2CO_2$ with different T doping concentrations at different H adsorption equivalent sites. When $|\Delta G_H| \leq 0.2$ eV, it indicates the material possess catalytic activity for HER.

Figure 3 shows the bond length, charge transfer, and first-order differential charge density in 2D-$Ti_3C_2O_2$ and 2D-$Ti_{27}C_{18}O_{17}X_1$. It was clear that the local structure became distorted and electronic structure space redistributed after being doped with X, which are main reasons for the improvement of HER activity. Taking 2D-$Ti_3C_2O_2$ and $Ti_{27}C_{18}O_{17}X_1$ as examples, the bond length of Ti-X gradually increased when O substituted by X, in detail, 1.969 Å for Ti-O, 2.394 Å for Ti-S, 2.532 Å for Ti-Se, and 2.756 Å for Ti-Te. With the increasing of bond length and the decreasing of electronegativity of Ti-X, the interaction between Ti and X was gradually weakened. From the analysis of charge transfer in Ti-X, the charges transferred from Ti to X gradually decreased from O to Te, and the charges transfer from O to O was $\Delta e = 1.109$ eV, S was $\Delta e = 0.836$ eV, Se was $\Delta e = 0.705$ eV, and that of Te was $\Delta e = 0.496$ eV, which is exactly opposite to the Ti-X bonds' length. In addition, the changes of local lattice structure and electronic structure caused by X doping are also displayed in the charge decomposition density diagram as shown in Figure 3 bottom images. By calculating and analyzing the deformation charge density, the properties of charge movement and polarization direction during bonding and bonding electron

coupling can be clearly obtained. First of all, bottom image in Figure 3a shows the change of electronic structure distribution of 2D-$Ti_3C_2O_2$ relative to isolated atoms. After being doped with X, there were obvious changes occurring at both distribution intensity and spatial configuration between Ti and X. It can be seen that the interaction strength from S to Te bonding with Ti decreased, which is consistent with the previous analysis of bond length and charge transfer.

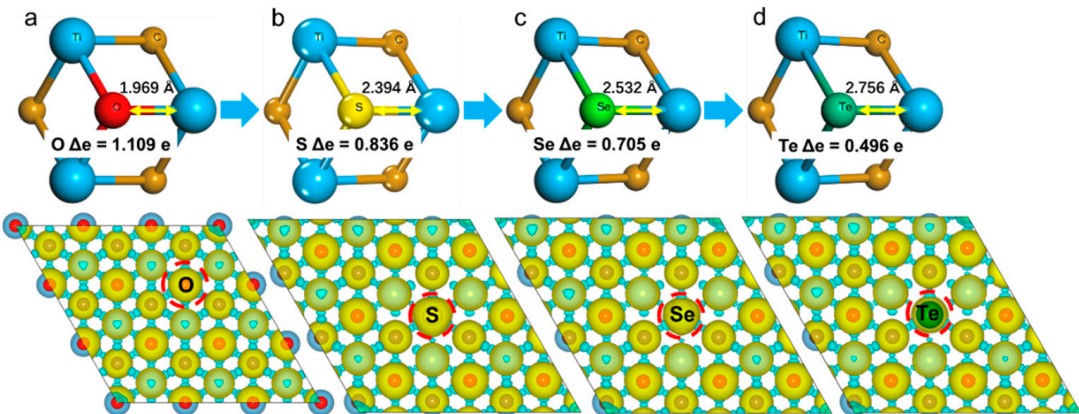

**Figure 3.** Bond length, charge transfer, and first-order differential charge density. (**a**) Top image shows the local crystal structure of the hydrogen adsorption site O of 2D-$Ti_3C_2O_2$, bond length, and charge transfer in O-Ti; bottom image shows the first-order deformation charge density diagram; (**b**–**d**) top images show the 2D-$Ti_{27}C_{18}O_{17}X_1$ local crystal structure of hydrogen adsorption site X, bond length, and charge transfer in Ti-X; the bottom images show the corresponding deformation charge density diagrams; (red dotted circles hinting the charge distribution of the active sites O or Te of HER catalyst).

The structural instability of 2D-$Ti_3C_2O_2$ after being doped with X can be seen from the local structural distortion caused by bonds length change of Ti-X. Electronegativity and atomic radius of X are two main factors for the change of bond length. The atomic radius enlarged and electronegativity decreased with the X atomic number increasing. The weakened and enlarged bonds length of Ti-X led to the decrease of structural stability. Furthermore, the phenomenon can also be verified by the charge transfer between X and the subsurface Ti. As shown in Figure 3, with the increasing of the local distortion between X and the subsurface Ti, the Bader charge transfer became smaller, which is obviously consistent with the increasing of local distortion and the decreasing of the charge transfer.

We calculated and added the spin density of 2D-$Ti_{27}C_{18}O_{18}$ before and after being doped with X in the supplementary material Figure S1. It was found that there were fewer spin down charges on the surface O atoms in the system $Ti_{27}C_{18}O_{18}$ without X doping. As shown in Figure S1b–d, after being X doped, there was obvious spin up charge in the doped element, which is obviously different from the surface O atom, and this will further affect the adsorption and catalytic activity of H.

Through doping X, the catalytic activity of 2D-$Ti_3C_2O_2$ was regulated by adjusting the local crystal structure and electronic structure around the doping position. Figure 4a,b show the electronic density of states analysis (DOS) of 2D-$Ti_{27}C_{18}O_{18}$ and $Ti_{27}C_{18}O_{17}Te_3$. It is shown that nonmetal doping caused the hybridization between p-orbitals of Te and d-orbitals of Ti, which led to the rearrangement of the electronic structures near the Fermi level. From −4 eV to Fermi level, the new p-orbital of Te was hybridized with the d-orbital of Ti not only changing the electronic total density of states analysis (TDOS), but also leading to the Fermi level increasing from −2.043 eV to −1.522 eV, and weakening the obtained electrons ability of $Ti_{27}C_{18}O_{17}Te_3$. Furthermore, the electronic density of states at Fermi level increased after being doped with Te, which is beneficial to electron conduction in 2D-$Ti_{27}C_{18}O_{18}$. After adsorbing H atoms, due to the hybridization of s-p orbitals between H-O and H-Te, a new peak was generated near −10 eV below the Fermi level as shown in Figure 4c,d. From insets of Figure 3c,d, we can see that hybridization degree

of O-H was significantly stronger than Te-H. In other words, the doping of Te changed the orbital electron hybridization, increased the Fermi energy level, and weakened the ability of capture electrons when interacting with H. As a result, the $\Delta G_H$ of 2D-$Ti_{27}C_{18}O_{18}$ decreased from $-2.193$ eV to $0.081$ eV for excellent HER catalytic activity.

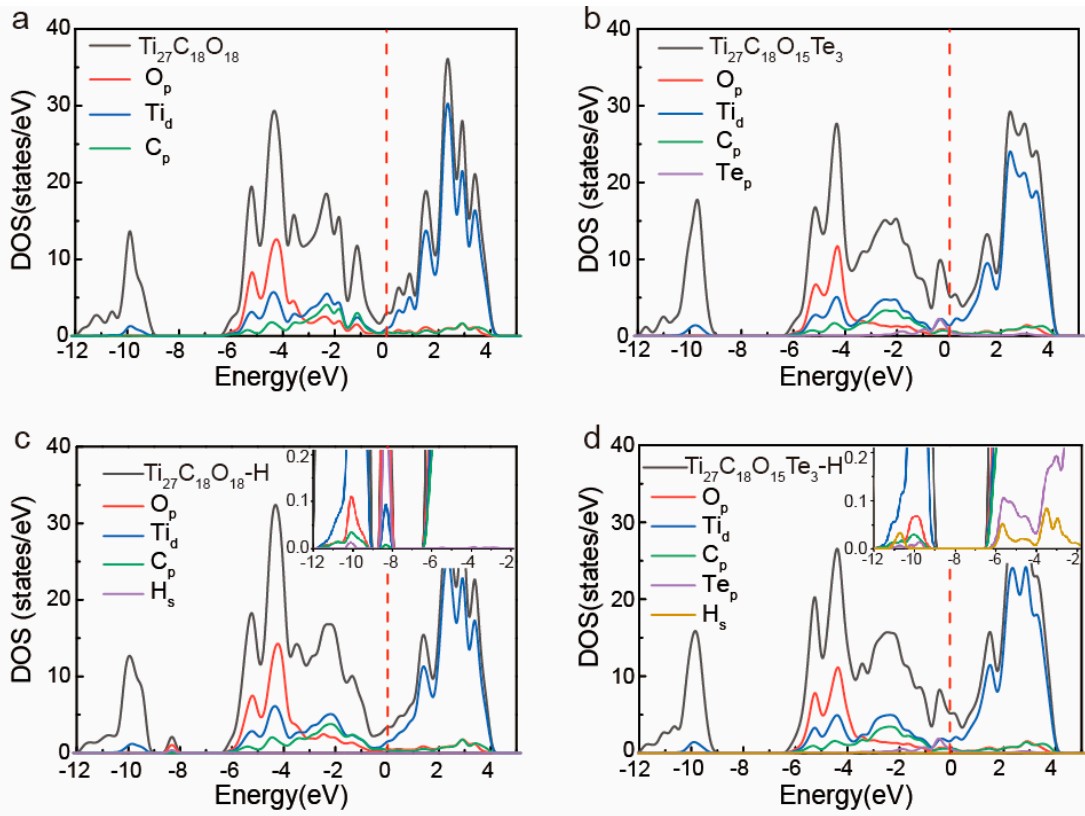

**Figure 4.** Density of states diagram. (**a,b**) total density of state (TDOS) and projected density of state (PDOS) of 2D-$Ti_{27}C_{18}O_{18}$ and $Ti_{27}C_{18}O_{17}Te_3$; (**c,d**) TDOS and PDOS of 2D-$Ti_{27}C_{18}O_{18}$ and $Ti_{27}C_{18}O_{17}Te_3$ after H adsorption; Fermi level is set to zero.

Figure 5 showed the charge density of 2D-$Ti_{27}C_{18}O_{18}$ before and after doped with X. Compared with pure 2D-$Ti_{27}C_{18}O_{18}$, the interaction between X and H was obviously weakened after being doped with X. Hydrogen atoms were adsorbed on the oxygen atoms' surface. The interaction between 2D-$Ti_{27}C_{18}O_{18}$ and H did not merely derive from the charge transfer between O and H on the surface, the Ti and C atoms in the structure also participated in the H adsorption due to the strong interaction. Moreover, Ti in the middle layer was also involved in the charge redistribution. In contrast, for $Ti_{27}C_{18}O_{17}Te_3$, the interaction with H was weakened due to the Fermi energy occurred shift after doped with Te. Because of the weak surface adsorption and asymmetric local environment, H atoms were preferred to be adsorbed on the surface, and only the surface atoms were involved in the adsorption, which verified the rationality of the previous local adsorption position and electronic structure analysis.

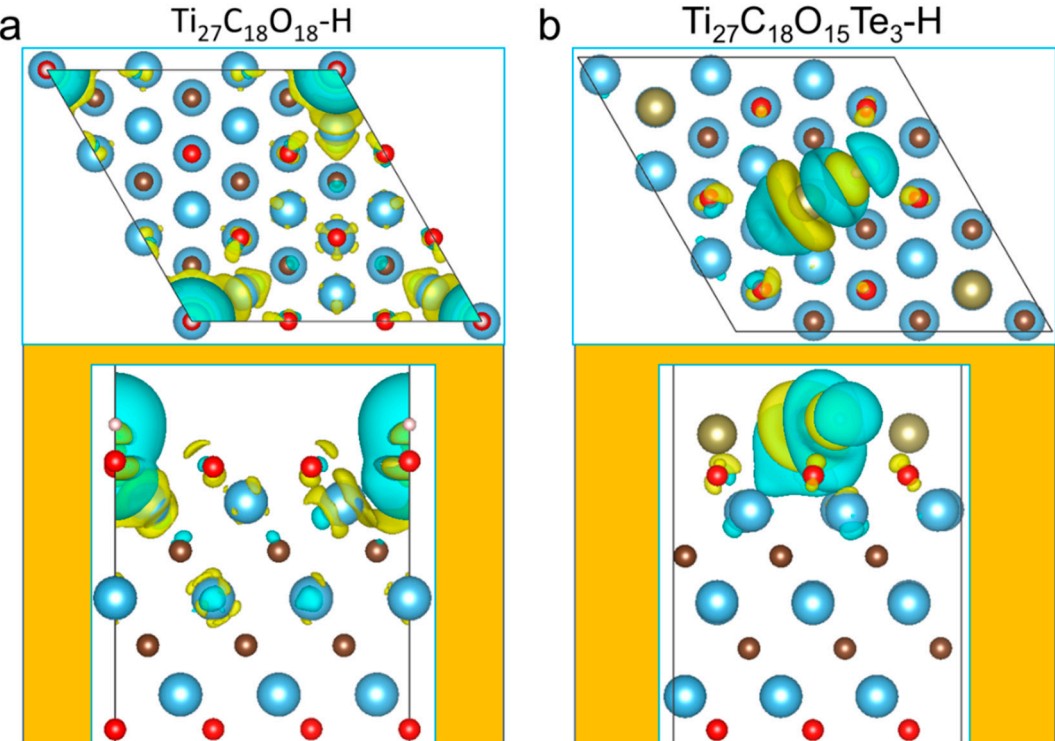

**Figure 5.** Difference charge density. (**a**) The different charge density maps of 2D-Ti$_2$CO$_2$ after H adsorption; (**b**) the different charge density maps of Ti$_{27}$C$_{18}$O$_{17}$Te$_3$ after H adsorbed.

Stability is an important indicator to evaluate the catalyst. To analyze the thermal stability of Ti$_{27}$C$_{18}$O$_{18}$ and Ti$_{27}$C$_{18}$O$_{17}$Te$_3$, we conducted an ab initio molecular dynamics (AIMD) study at 300 K of canonical ensemble (NVT ensemble) as shown in Figure 6. Both 2D-Ti$_{27}$C$_{18}$O$_{18}$ and Ti$_{27}$C$_{18}$O$_{17}$Te$_3$ showed stable structure and steady temperature fluctuation after running 1200 fs. Although the local structure changed slightly, the overall morphology remained good. The result shows that 2D-Ti$_{27}$C$_{18}$O$_{18}$ and Ti$_{27}$C$_{18}$O$_{17}$Te$_3$ can stably exist at room temperature, which is conducive to the experimental synthesis.

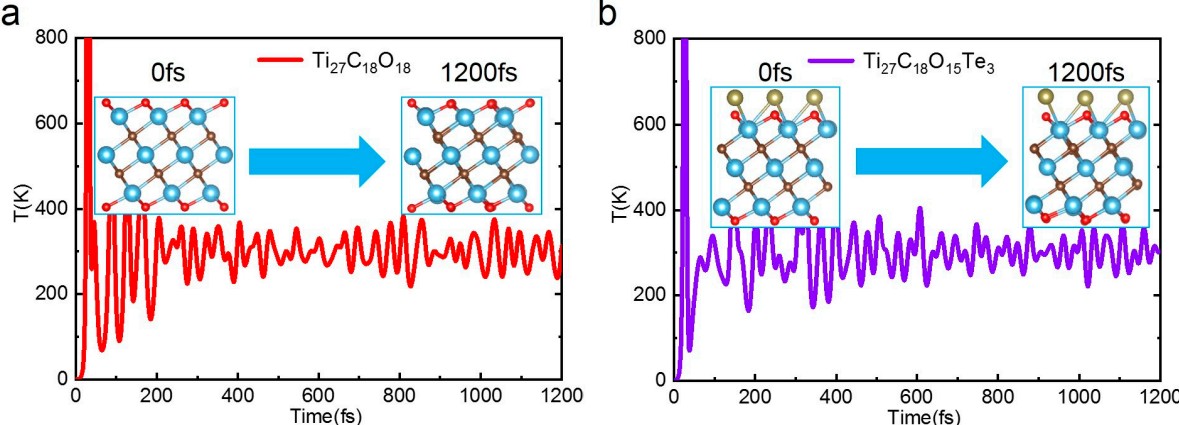

**Figure 6.** Ab initio molecular dynamics (AIMD) verification: (**a**) AIMD of 2D-Ti$_2$CO$_2$ at 300 K of canonical ensemble (NVT ensemble); (**b**) AIMD of Ti$_{27}$C$_{18}$O$_{17}$Te$_3$ at 300K of NVT ensemble; the illustrated structure shows the initial and final structural sketch.

### 3. Computational Methodology

Spin-polarized density functional theory (DFT) was conducted in the Vienna ab initio simulation package (version vasp5.3, 2015) [18,19]. We adopted potentials in the projected augmented-wave method (PAW) and the exchange-correlation energy in the general gradient approximation (GGA) in the scheme proposed by Perdew–Burke–Ernzerh (PBE) [20,21]. The cut-off kinetic energies for the plane waves were set to 450 eV for all the calculations. The convergence tolerance of energy and force on each atom during structure relaxation was less than $10^{-5}$ eV and 0.02 eV/Å. A set of Monkhorst–Pack mesh K points of $9 \times 9 \times 1$ and $11 \times 11 \times 1$ were used to sample the Brillouin zone for geometry optimization and electronic structural calculations. A vacuum layer of 20 Å was set up to avoid interactions between two periods. The semiempirical dispersion-corrected DFT force-field approach (DFT-D3) was adopted to describe the weak interaction involved in the calculations [22]. DFT + U calculation should be considered in transition metal compounds considering the strong correlation between electrons. Previous studies demonstrated that + U does not affect the catalytic activity of 2D-$Ti_3C_2O_2$. Therefore, in our work, U was not adopted [23–25]. Ab initio molecular dynamics (AIMD) simulations containing large-enough super-cells were used to evaluate the thermal stability of 2D-$Ti_3C_2O_2$ after S, Se, and Te doped structures [26]. The systems were stabilized at 300 K for 1.2 ps with a time step of 1 fs, where the algorithm of Nosè was used to control the temperature.

To evaluate the electrocatalytic HER activity, the Gibbs free energy of hydrogen adsorption $\Delta G_H$ was a key descriptor and defined as [4,24]

$$\Delta G_{H*} = \Delta E_H + \Delta E_{ZPE} - T\Delta S_H \tag{1}$$

where $\Delta E_H$ is the adsorption energy for adding one H atom onto the $Ti_2CO_2$ catalysts.

$$\Delta E_H = E_{H*} - E - 1/2E_{H_2} \tag{2}$$

where $\Delta E_{H*}$, $\Delta E$ and $E_{H_2}$ are total energies of the catalyst with adsorbed H atom, the catalyst without adsorbed H atom, and $H_2$ for gas phase, respectively. $\Delta E_{ZPE}$ and $T\Delta S_H$ are the zero-point energy and entropy differences between the adsorbed system and gas phase, respectively. $\Delta E_{ZPE}$ can be calculated using Equation

$$\Delta E_{ZPE} = E_{ZEP}^{H*} - E_{ZPE} - 1/2E_{ZEP}^{H_2} \tag{3}$$

where $E_{ZPE}^{H*}$ represent zero-point energy with a H atom adsorbed on the surface of catalyst, $E_{ZPE}^{H_2}$ represents zero-point energy of isolate $H_2$ molecule.

The $\Delta S_H$ can be approximated as

$$\Delta S_H \approx 1/2S_{H_2}^0 \tag{4}$$

due to the fact that the vibrational entropy in the adsorbed state is small according to previous studies [24], and $S_{H_2}^0$ is the entropy of $H_2$ gas under the standard conditions [21]. Here, the values of $\Delta E_{ZPE}$ and $T\Delta S_H$ are referenced from Ref. [8]. Therefore, Equation (5) can be written as [25]

$$\Delta G_{H*} = \Delta E_H + 0.3 \, eV \tag{5}$$

$\Delta G_H$ as a descriptor of catalytic activity trend is a necessary and insufficient condition. If other conditions remain unchanged, when $\Delta G_H$ approaches 0, the potential barrier for adsorption and desorption is smaller and the corresponding catalytic activity is better. Therefore, we used $\Delta G_H$ as a descriptor to study the regulation of nonmetal doping on the catalytic hydrogen evolution of 2D MXenes [23,24,27] The optimal $\Delta G_H$ value for HER is close to 0 eV, which means that the smaller the value is $|\Delta G_H|$, the better the HER performance of the catalyst will be.

## 4. Conclusions

In summary, we studied the HER catalytic activity of 2D-$Ti_3C_2O_2$ after being doped with X by density functional theory. The calculation of $\Delta G_H$ shows that X doping can effectively promote the catalytic activity by changing the bond length and electrical structure of the adsorption sites. $Ti_{27}C_{18}O_{17}Te_3$ showed the best catalytic activity for HER. The hybridization of p-d orbitals led to the shift of Fermi level and weakened the interaction with H, which was verified by DOS and difference charge density. Moreover, AIMD analysis showed that the doped system maintained stability, which forebode the materials can be successfully prepared in experiment. This work not only screens out the 2D-$Ti_{27}C_{18}O_{17}Te_3$ as high-performance catalyst for HER from the theoretical study, but also provides theoretical guidance for the design of other 2D-MXenes electrocatalytic HER materials and their successful synthesis in experiment.

**Supplementary Materials:** The following are available online at https://www.mdpi.com/2073-4344/11/2/161/s1. Table S1. Formation Energy ($E_f$) of 2D-$Ti_3C_2O_2$ before and after doped with X (X = S, Se, Te) in the 3 × 3 supercell. Bond Length ($B_{NM}$-Ti) of Ti-X, charge transfer (Bader) of X. Positive values denude the obtained electron and the corresponding lattice constant (a). Table S2. Structure optimization results of 2D-$Ti_3C_2O_2$. Figure S1. 3 × 3 supercell spin density of 2D-$Ti_3C_2O_2$ before and after doped with X. (a) $Ti_{27}C_{18}O_{18}$; (b) $Ti_{27}C_{18}O_{15}S_1$; (c) $Ti_{27}C_{18}O_{15}Se_1$; (d) $Ti_{27}C_{18}O_{15}Te_1$. Yellow: Spin up electrons, Blue: Spin down electrons.

**Author Contributions:** F.L.: data processing and analysis, writing—original draft; X.W.: performed calculations, data collection and analysis; R.W.: project administration, resources, supervision, validation, funding acquisition. All authors have read and agreed to the published version of the manuscript.

**Funding:** This study was supported by the National Key Research and Development Program of China (No. 2018YFA0703700), the National Natural Science Foundation of China (Nos. 51971025 and 51901012), the Fundamental Research Funds for the Central Universities (FRF-TP-17-073A1) and 111 Project (No. B170003).

**Data Availability Statement:** The data presented in this study are openly available in https://digital.csic.es/.

**Conflicts of Interest:** The authors declare no conflict of interest.

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
