# Peer review of "Surface Modifications of 2D-Ti3C2O2 by Nonmetal Doping for Obtaining High Hydrogen Evolution Reaction Activity: A Computational Approach"

_catalysts, doi:10.3390/catal11020161_

Round 1

Reviewer 1 Report

The manuscript ” Surface Modifications of 2D-Ti3C2O2 by Nonmetal  Doping for Obtaining High Hydrogen Evolution Reaction Activity: A Computational Approach” by Wang et al. deals with the computational investigation of the doping of the 2D-Ti3C2O2 material with different chalcogen atoms. The article seems interesting since it provides some useful hints for the optimization of the conditions for the HER activity of the material. In any case before acceptance, the manuscript has to be modified by taking into account different points:

  • First of all, structural/electronic correlations have to be described in the manuscript. What about the structural deformation of the material by the different doping degree? I wonder that as the doping becomes more consistent the structure should be particularly deformed, especially with the heavier chalcogen atoms. The present reviewer did not find any hints nor in the SI. How these deformations could be related with the electronic structure of the doped surface? The authors must describe clearly these correlations.
  • Figure 1 has to be changed since it is not clear for a reader. In the text, the authors report “are three kinds of H adsorption sites T, O1 and O2” but this is must be more evident from the Figure.
  • For the sake of clarity, the authors have to include in the Supporting Information the coordinates of the all optimized structures together with the obtained related energy features.
  • The manuscript should be carefully re-read and corrected by a native speaker, since there are several language related problems.

According to the present reviewer, all the aforementioned points have to be addressed by the authors before the reconsideration and the acceptance of the manuscript.   

Author Response

The manuscript “Surface Modifications of 2D-Ti3C2O2 by Nonmetal Doping for Obtaining High Hydrogen Evolution Reaction Activity: A Computational Approach” by Wang et al. deals with the computational investigation of the doping of the 2D-Ti3C2Omaterial with different chalcogen atoms. The article seems interesting since it provides some useful hints for the optimization of the conditions for the HER activity of the material. In any case before acceptance, the manuscript has to be modified by taking into account different points:

  1. First of all, structural/electronic correlations have to be described in the manuscript. What about the structural deformation of the material by the different doping degree? I wonder that as the doping becomes more consistent the structure should be particularly deformed, especially with the heavier chalcogen atoms. The present reviewer did not find any hints nor in the SI. How these deformations could be related with the electronic structure of the doped surface? The authors must describe clearly these correlations.

3x3 supercell

Ef (eV)

BNM-Ti (Å)

Bader (eV)

a (Å)

Ti27C18O18

-7.454

1.969

1.109

9.059

Ti27C18O17S1

-7.4204

2.394

0.836

9.065

Ti27C18O17S3

-7.349

2.390

0.824

9.107

Ti27C18O17Se1

-7.409

2.532

0.705

9.067

Ti27C18O17Se3

-7.315

2.532

0.700

9.109

Ti27C18O17Te1

-7.393

2.756

0.496

9.082

Ti27C18O17Te3

-7.268

2.770

0.490

9.121

Table S1 Formation Energy (Ef) of 2D-Ti3C2O2 before and after doped with X (X=S, Se, Te) in the 3×3 supercell. Bond Length (BNM-Ti) of Ti-X, charge transfer (Bader) of X. Positive values denude the obtained electron and the corresponding lattice constant (a).

Response: Thanks for your reviewing! Actually, elements doping can lead to structural distortion, and the distortion extent is increasing with the doping elements’ radius enlarging, meanwhile, the structure becoming unstable. As shown in Figure 3, we can see that the bonds length of Ti-X is depend on the atoms’ radius, and there is obvious local structure distortion occurred, which leads to the relative instability of the structure. According to the formation energy (Table S1), it also be demonstrated that the structure energy of 2D-Ti3C2O2 is increasing with the doped atomics’ radius enlarging, and the system becoming unstable. However, the negative formation energy indicating 2D-Ti3C2O2 possess thermodynamic stability. Therefore, based on the first principle molecular dynamics, we verified that Ti27C18O17Te3 can maintain a good morphology and stability at room temperature. We have added the following description in the main manuscript “The structural instability of 2D-Ti3C2O2 after doped with X can be reflected from the local structural distortion caused by bonds length change of Ti-X. Electronegativity and atomic radius of X are two main factors for the change of bond length. The atomic radius is enlarging and electronegativity is decreasing with the X atomic number increasing. The weakened and enlarged bonds length of Ti-X leads to the decrease of structural stability. Furthermore, the phenomenon also can be verified by the charge transfer between X and the subsurface Ti. As shown in Figure 3, with the increasing of the local distortion between X and the subsurface Ti, the Bader charge transfer becoming smaller, which is obviously consistent with the increasing of local distortion and the decreasing of the charge transfer.

  1. Figure 1 has to be changed since it is not clear for a reader. In the text, the authors report “are three kinds of H adsorption sites T, O1 and O2” but this is must be more evident from the Figure.

Figure 1. Structure diagram and ΔGH. (a) 2D-Ti2CO2 (Ti27C18O18) of 3×3 supercell and (b)Ti27C18O17X and Ti27C18O17X3 doped with different concentrations of X (X=S, Se, and Te)and (c) corresponding H equivalent adsorption sites X, O1, and O2.

Response: Thanks for your careful reviewing! Before X doping, there is only one equivalent position on the surface for H adsorption, named O1.When doped one X, the equivalent positions are increased to three, named O1, O2 and X, respectively. When doped three X, there are two equivalent positions on the surface for H absorption, named O1 and X. We have marked the three kinds of H adsorption sites O1, O2, and X in the reprocessed figure 1(c).

  1. For the sake of clarity, the authors have to include in the Supporting Information the coordinates of the all optimized structures together with the obtained related energy features.

3×3 supercell

Ef (eV)

BNM-Ti (Å)

Bader (eV)

a (Å)

Ti27C18O18

-7.454

1.969

1.109

9.059

Ti27C18O17S1

-7.4204

2.394

0.836

9.065

Ti27C18O17S3

-7.349

2.390

0.824

9.107

Ti27C18O17Se1

-7.409

2.532

0.705

9.067

Ti27C18O17Se3

-7.315

2.532

0.700

9.109

Ti27C18O17Te1

-7.393

2.756

0.496

9.082

Ti27C18O17Te3

-7.268

2.770

0.490

9.121

Table S1 Formation Energy (Ef) of 2D-Ti3C2O2 before and after doped with X (X=S, Se, Te) in the 3×3 supercell. Bond Length (BNM-Ti) of Ti-X, charge transfer (Bader) of X. Positive values denude the obtained electron and the corresponding lattice constant (a).

Response: Thanks for your reviewing! We have added the Formation Energy (Ef) of 2D-Ti3C2O2 before and after doped with X in the 3×3 supercell as shown in table S1. Bond Length (BNM-Ti) of Ti-X, charge transfer (Bader) of X. Positive values denude the obtained electron and the corresponding lattice constant (a) in the supplementary materials.

  1. The manuscript should be carefully re-read and corrected by a native speaker, since there are several language related problems.

Response: Thanks for your careful reviewing! We have carefully corrected the language mistakes and improved the quality of the revised manuscript.

Reviewer 2 Report

In this computational work, the authors have discussed the catalytic ability of the 2D-Ti3C2O2 materials by calculating the hydrogen adsorption free energy with the VASP-DFT code. In particular, they have focused on the effect of mon-metal (S, Se, and Te) doping on the catalytic ability. Basically, this research is interesting; however, some points including major ones should be addressed before the manuscript is to be published.

  1. Since no experimental data, that can be compared to this computational results, is available, I think that the authors should be careful about the accuracy (reliability) of the obtained result. They do not provide any comments on the accuracy of the computational method at all. It is very dangerous to derive conclusions without benchmarking calculations or comparisons to other works. More qualitative analyses as well as quantitative discussion should be given. For example, why is the PBE functional used (I do not think that this functional yields the most accurate results)?
  2. I do not understand why the authors are reporting constant-temperature MD results. I do not understand that the MD calculations were done only for limited samples. In my understanding, geometry optimization was done in all the cases (is this correct?) in this study. The stability of materials can be discussed in terms of the optimized properties including vibrational frequencies. Of course, phase transitions may be important in some cases.
  3. Please add more detailed discussion on the relation between ΔGH and the doped atom. I do not understand what kind of atomic properties are important for the absolute values of ΔGH.
  4. Is it possible to report potential energy curves for the hydrogen adsorption processes?
  5. Probably, Eq. (5) is wrong.
  6. I strongly recommend the authors to use other characters such as “X” instead of “T” to define the doped atom. “T” generally means tritium and is not appropriate.

Author Response

In this computational work, the authors have discussed the catalytic ability of the 2D-Ti3C2O2 materials by calculating the hydrogen adsorption free energy with the VASP-DFT code. In particular, they have focused on the effect of mon-metal (S, Se, and Te) doping on the catalytic ability. Basically, this research is interesting; however, some points including major ones should be addressed before the manuscript is to be published.

  1. Since no experimental data that can be compared to this computational results, is available, I think that the authors should be careful about the accuracy (reliability) of the obtained result. They do not provide any comments on the accuracy of the computational method at all. It is very dangerous to derive conclusions without benchmarking calculations or comparisons to other works. More qualitative analyses as well as quantitative discussion should be given. For example, why is the PBE functional used (I do not think that this functional yields the most accurate results)?

Ti3C2O2

a (Å)

O-Ti1 bond (Å)

Ti1-C bond(Å)

C-Ti2 bond(Å)

This work

3.020

1.969

2.179

2.144

Reference

3.019

1.970

2.180

2.144

Table S2 Structure optimization results of 2D-Ti3C2O2.

Response: Thanks for your careful reviewing! Our calculation method and parameters are similar as the previous reported reference (J. mater. Chem. A, 2020, 8, 23488-23497). The calculated crystal structure parameters were also be compared as shown in Table S2. It is found that the calculated results are consistent with the previous reference. These comparisons have been added in the Table S2 of supplementary materials.

  1. I do not understand why the authors are reporting constant-temperature MD results. I do not understand that the MD calculations were done only for limited samples. In my understanding, geometry optimization was done in all the cases (is this correct?) in this study. The stability of materials can be discussed in terms of the optimized properties including vibrational frequencies. Of course, phase transitions may be important in some cases.

Response: Thanks for your careful reviewing! All the structures have been optimized, which can be determined to be stable by structural parameters and formation analysis. Since the catalyst is usually used at/above room temperature, thus, we used AIMD to test the thermodynamic stability of Te doped 2D-Ti3C2O2 at 300K. This method have been applied in many works, such as references (J. mater. Chem. A, 2020, 8, 23488-23497)

  1. Please add more detailed discussion on the relation between ΔGHand the doped atom. I do not understand what kind of atomic properties are important for the absolute values of ΔGH.

Response: Thanks for your careful reviewing! “We have added the principle of hydrogen atom adsorption on nonmetallic elements (Fig. 2a) and the relationship between the catalytic activity and Bader charge transfer of doped X (Fig. 2b). It was found that the decreasing electronegativity of X result in the ability of 2D-Ti3C2O2 for obtain charge (Bader charge transfer) is decreasing after doped with X. From figure. 2b, we can see that relationship between Bader charge and catalytic activity (ΔGH) presented inverse proportional, and the ΔGH value is positive correlation with the doping concentration of X. The intrinsic relationship between the catalytic activity and the doping elements provides a theoretical basis for the design and synthesis of new MXenes catalysts for hydrogen evolution.” We have added the above description in the revised manuscript.

  1. Is it possible to report potential energy curves for the hydrogen adsorption processes?

Response: Thanks for your careful reviewing! The process of electrocatalytic hydrogen evolution can be ascribe to following two steps: adsorption and desorption, although the details of hydrogen adsorption can be reflected by the adsorption energy curve. However, only H * (* denotes adsorption) is the intermediate product of hydrogen evolution reaction, and the tendency of catalytic activity can be reflected by the Gibbs free energy ΔGH of hydrogen adsorption as a descriptor. Therefore, the regulatory effect of different kinds and concentrations of nonmetal doping on ΔGH has been calculated in this paper, and the potential energy curve of hydrogen adsorption process is not further calculated.

  1. Probably, Eq. (5) is wrong.

Response: Thanks for your careful reviewing! We have corrected the Eq. (5) as “∆G_(H*)=∆E_H+0.3 eV”.

  1. I strongly recommend the authors to use other characters such as “X” instead of “T” to define the doped atom. “T” generally means tritium and is not appropriate.

Response: Thanks for your recommend, we have used “X” to define the doped atom S, Se and Te.

Reviewer 3 Report

Review comment for the article “Surface Modifications of 2D-Ti3C2O2 by Nonmetal Doping for Obtaining High Hydrogen Evolution Reaction: A Computational Approach”

              The authors suggested the effects of nonmetal atoms (S, Se, and Te) doping on the HER activity of Ti3C2O2 via the DFT calculations. The authors should clarify the following unclear point to publish the article.

              The author argued that more negative delta_GH will result in a too strong bonding of adsorbed H to extract from the catalyst surface, while more positive delta_GH will make the protons bond to the surface of catalyst too weak and difficult, both leading to slow HER kinetics. This indicates that there is an optimal value of delta_GH for the HER, i.e. we can investigate the HER by the volcano plot. I agree it. However, I disagree the optimal value (the top of volcano) is the “delta_GH = 0”. The authors should show strict proof why the optimal value become zero.

In addition,

  1. I recommend that the authors show the spin densities of Ti27C18O18, Ti27C18O17S1, Ti27C18O15S3, Ti27C18O15Se1, Ti27C18O15Se3, Ti27C18O17Te1, and Ti27C18O15Te3. The spin density will correspond to the unpaired electron densities, and we can confirm the position of the dangling bond that will be important for H adsorption.

  1. In equations (2) and (3), there are energies whose definitions are not shown.

  1. In equation (5), S is G? (Is it typo?)

  1. Why is the reference [10] highlighted by red colour?

  1. As referred the mechanism argued by the authors, the Ti 3d orbital will be important. Hence, the author should perform DFT+U calculations and should confirm the U dependency.

Author Response

Review comment for the article “Surface Modifications of 2D-Ti3C2O2 by Nonmetal Doping for Obtaining High Hydrogen Evolution Reaction: A Computational Approach” The authors suggested the effects of nonmetal atoms (S, Se, and Te) doping on the HER activity of Ti3C2O2 via the DFT calculations. The authors should clarify the following unclear point to publish the article.The author argued that more negative delta_GH will result in a too strong bonding of adsorbed H to extract from the catalyst surface, while more positive delta_GH will make the protons bond to the surface of catalyst too weak and difficult, both leading to slow HER kinetics. This indicates that there is an optimal value of delta_GH for the HER, i.e. we can investigate the HER by the volcano plot. I agree it. However, I disagree the optimal value (the top of volcano) is the “delta_GH = 0”. The authors should show strict proof why the optimal value become zero.
Response: Thanks for your careful reviewing! As mentioned in the literature (Z. W. sEH et al., science 355, eaad4998, 2017), ΔGH as a descriptor of catalytic activity trend is only a necessary and insufficient condition. If other conditions remain unchanged, when ΔGH approaches 0, the potential barrier for adsorption and desorption is smaller and the corresponding catalytic activity is better. The detailed description of this theory is described in the literature (J. K. n ü rskov et al 2005, J. electrochem. SOC. 152 J23). It has been widely used to predict the catalytic hydrogen evolution reaction by using ΔGH as a descriptor (J. mater. Chem. A, 2020, 8, 23488-23497). Therefore, we also used ΔGH as a descriptor to study the regulation of nonmetal doping on the catalytic hydrogen evolution of 2D MXenes.
1. I recommend that the authors show the spin densities of Ti27C18O18, Ti27C18O17S1, Ti27C18O15S3, Ti27C18O15Se1, Ti27C18O15Se3, Ti27C18O17Te1, and Ti27C18O15Te3. The spin density will correspond to the unpaired electron densities, and we can confirm the position of the dangling bond that will be important for H adsorption.

Figure S1 3×3 supercell spin density of 2D-Ti3C2O2 before and after doped with X. (a)Ti27C18O18; (b) Ti27C18O15S1; (c) Ti27C18O15Se1; (d) Ti27C18O15Te1. Yellow: Spin up electrons, Blue: Spin down electrons.
Response: Thanks for your careful reviewing! We have calculated and added the spin density of Ti27C18O18 before and after doped with X in the supplementary material figure S1. It was found that there are less spin down charges on the surface O atoms in the system Ti27C18O18 without X doping., As shown in figure S1 (b-d), after X doped, there is obvious spin up charge in the doped element, which is obviously different from the surface O atom, and this will further affect the adsorption and catalytic activity of H. 
2. In equations (2) and (3), there are energies whose definitions are not shown.
Response: Thanks for your careful reviewing! For equation (2), ∆E_H=E_(H*)-E_slab-1/2E_H2 where E_(H*)  ,E_slab  and〖 E〗_H2  are total energies of the catalyst with adsorbed H atom, the catalyst without adsorbed H atom, and H2 for gas phase, respectively.
For equation (3) ∆E_ZPE=E_ZEP^(H*)-1/2E_ZEP^H2 where E_ZPE^(H*) represent zero-point energy with a H atom adsorbed on the surface of catalyst, E_ZPE^(H_2 ) represents zero-point energy of isolate H2 molecule.
3. In equation (5), S is G? (Is it typo?)
Response: Thanks for your careful reviewing! We have corrected equation (5) as “∆G_(H*)=∆E_H+0.3 eV”.
4. Why is the reference [10] highlighted by red colour?
Response: Thanks for your careful reviewing! The red colour is meaningless, we have cleared it.
5. As referred the mechanism argued by the authors, the Ti 3d orbital will be important. Hence, the author should perform DFT+U calculations and should confirm the U dependency.
Response: Thanks for your careful reviewing! DFT + U calculation should be considered in transition metal compounds considering the strong correlation between electrons. However, in the 2D-Ti3C2O2 system, our previous work (J. mater. Chem. A, 2020, 8, 23488-23497) and other references (J. catalyst, 387,2020) have verified that + U does not affect the catalytic activity trend. So, in this work U is not be adopted, which is also commonly used in the theoretical study on 2D Mxenes materials.

Round 2

Reviewer 1 Report

The authors have done appropriate revisions based on the points raised by the present reviewer. According to me, the manuscript can be publishable in the present form.

Author Response

Thanks very much for your contribution to our manuscript!

Reviewer 2 Report

I am very pleased since the authors have made appropriate revisions based on the points raised by the referee. I therefore conclude that the manuscript can now be published as it is.

Author Response

(The authors gave the same response as above.)

Reviewer 3 Report

The authors took a effort to improve their study, and their responses to the referee's comment was good. However, some responses were not used for the revisions. Namely, the reason of "GH=0 is good catalysts" and "DFT+U is not needed for the discussions on the Ti3C2O2" were not described in the manuscript. The authors investigated these points in their previous works. Please add the sentences of (based on) the responses with the certain references.

Author Response

Reviewer #3

The authors took a effort to improve their study, and their responses to the referee's comment was good. However, some responses were not used for the revisions. Namely, the reason of "GH=0 is good catalysts" and "DFT+U is not needed for the discussions on the Ti3C2O2" were not described in the manuscript. The authors investigated these points in their previous works. Please add the sentences of (based on) the responses with the certain references.

Response: Thanks for your careful reviewing! We have added the following statement in the revised manuscript to explain “=0 is good catalysts” and “DFT+U is not needed for the discussions on the Ti3C2O2”. “ as a descriptor of catalytic activity trend is a necessary and insufficient condition. If other conditions remain unchanged, when  approaches 0, the potential barrier for adsorption and desorption is smaller and the corresponding catalytic activity is better. Therefore, we used as a descriptor to study the regulation of nonmetal doping on the catalytic hydrogen evolution of 2D Mxenes. [23, 24, 27].“DFT + U calculation should be considered in transition metal compounds considering the strong correlation between electrons. Previous studies demonstrated that + U hasn’t affect the catalytic activity of 2D-Ti3C2O2. So, in our work U is not be adopted. [23-25]”
